# Water Use Efficiency of Soybean under Water Stress in Different Eroded Soils

**Shuang Li [1], Yun Xie [1,\*], Gang Liu [1], Jing Wang [2], Honghong Lin [1], Yan Xin [1] and Junrui Zhai [1]**

1   State Key Laboratory of Earth Surface Processes and Resource Ecology, Faculty of Geographical Science, Beijing Normal University, Xinjiekouwai Str.19, Beijing 100875, China; shuangli@mail.bnu.edu.cn (S.L.); liugang@bnu.edu.cn (G.L.); lhh@mail.bnu.edu.cn (H.L.); xinyanhao51@163.com (Y.X.); 18734804826@163.com (J.Z.)
2   Resources and Environment Economy College, Inner Mongolia University of Finance and Economics, Hohhot 010070, China; 201631170007@mail.bnu.edu.cn
\*   Correspondence: xieyun@bnu.edu.cn

**Abstract:** Soil erosion could change the effective storage of soil moisture and affected crop water use efficiency (WUE). To quantitative study differences in the WUE of soybean and the crop's response to water stress for soils with different degrees of erosion in northeastern China, three erosion degrees—(1) lightly, (2) moderately and (3) severely—eroded black undisturbed soils and four years (from 2013 to 2016) of soybean pot experiments were used to control soil water content (100%, 80%, 60%, and 40% field capacity (FC)) and observe the crop growth processes. To study the relationships between erosion–water use–productivity, the following results were achieved: (1) the optimal water content was 80% FC for lightly eroded soil (L) and 100% FC for both moderately (M) and severely (S) eroded soil. Yield (Y) was best in M with the value of 3.12 t ha$^{-1}$, which was 4.6% and 85.5% higher than L and S, respectively. Under the conditions of adequate water supply, there was no significant change in Land M, but the values were significantly different for the S ($p < 0.05$). (2) Y and biomass (B) were sensitive to water stress except in the branching stage. (3) The values of WUE$_Y$ and WUE$_B$ for the three eroded soils were the best at 80% FC. The stress coefficient (SF) values of the three eroded soils were not significantly different. In the flowering and pod formation stage, the SF reached the maximum under waterlogging stress. While the water shortage stress reached the maximum in the seed filling stage, the soil water content decreased by 10%, and the WUE$_B$ decreased by 15%, which was 2.5 times more powerful than the waterlogging stress. This study indicated the change in soybean growth with respect to the water response caused by soil erosion, and provided a scientific basis and data for the reasonable utilization of black soil with different erosion intensities. The results also provided important parameters for the growth of simulated crops.

**Keywords:** soil erosion; soybean growth; water stress; water use efficiency

## 1. Introduction

Water use efficiency (WUE) is affected by soil erosion [1–3]. The effect mainly manifests in the following three aspects. First, soil erosion leads to a change in soil's physicochemical properties and structure [4–7], and these variations cause differences in aeration of the rhizosphere [8,9]; the benefit of aeration is more pronounced with increasing WUE [10]. Second, soil fertility and organic matter are changed by soil erosion [11–14]. Fertility has a great potential to further increase of the efficiency of water use [15–18]. N is important in improving WUE and soil water use [19,20], while P plays an important role in increasing not only total soil water use but also water extraction from deep soil layer [21,22]. Third, since soil erosion increases runoff, and reduces water infiltration [23] as well as

soil's effective capacity, and eventually influences the soil water storage [24–33], a poor water-holding capacity increases the drainage of rainwater and irrigation water below the root zone, resulting in low water use efficiency (WUE) of crops [34].

As one of the most important grain production bases, the black soil region located in Northeast China was reclaimed only approximately 100 years ago but has experienced serious soil erosion [35] under the spring freezing and thawing effect and heavy storms occurring in the rainy season [36,37]. Meanwhile, conventional tillage has caused the loss of soil organic carbon and the severe soil degradation of soil structure in this region [38]. The moderately and severely water-eroded area accounted for 31.4% and 7.9% of the total [38]. Since the 1950s, the average black soil thickness has decreased by 40 cm [39]. In this region, erosion–water use–productivity relationships have attracted increasing research attention, such as Li and Sun [40], using a model to simulate the effects of supplementary irrigation on yield, water use efficiency (WUE) and irrigation water use efficiency (IWUE) on corn. Their results showed that single irrigation increased corn yields by 3%–35% for aeolian sandy soil and 5%–35% for black soil compared with rain-fed conditions based on 33 year weather data. Yang et al. [41], modeling the effects of plastic film mulching on irrigated maize yield and water use efficiency in sub-humid Northeast China, found the average simulated grain yield and WUE for mulched treatments were 8% and 13% greater compared to no-mulched. Wu et al. [42] analyzed the effect of different drip fertigation methods on maize yield, nutrient and water productivity in two soils (clay and sandy soil) in Northeast China. The result showed that drip fertigation improved soil water and increased maize yield and water use efficiency. Among those studies, the effects of erosion on the processes of crop growth and WUE are rarely stated, and many studies focused on the crops are maize [38,43–47], and wheat [46].

Soybean as a native of China and one of the most important crops in the country, has been known to man for over 5000 years [48], and 56% of China's total soybean production was provided in northeastern China [49]. The agriculture in northeastern China is critically dependent on rainfall and water use efficiency [16,18,50], and rainfed crop production is subject to frequent fluctuations in precipitation [51]. Thus, in northeast China, a quantitative analysis of the relationship between WUE and water supply is extraordinarily important during the growth process of the crop.

In this study, three erosion degrees, (1) lightly, (2) moderately and (3) severely eroded black undisturbed soils, were collected, to study the relationships between erosion–water use–productivity. Four years (from 2013 to 2016) of soybean pot experiments were used to control soil water content (100%, 80%, 60%, and 40% field capacity (FC)) and observe the crop growth processes. The objectives of this study were to (1) confirm the optimal soil moisture conditions for crop growth indicators in different eroded soils, and the differences under optimal water supply conditions; (2) analyze the sensitivity of soybean yield and biomass to water; (3) quantitative analysis of the relationship between WUE and water stress for crop growth processes in three erosion soils.

## 2. Materials and Methods

### 2.1. Study Area

The study was conducted at the Jiusan soil and Water Conservation Experimental Station of Beijing Normal University, which is located in the Heshan Farm (northwestern part of Heilongjiang Province, 125°16′ E–125°21′ E, 48°59′ N–49°03′ N). Elevation ranges from 310 to 390 m, with long and gentle slopes. The slope length is up to 800–1500 m and the slope angle is generally between 1° and 4° [35,52]. The climate is the mid-temperate continental monsoon climate, with an annual average temperature of 0 °C and a large temperature difference between winter and summer. Annual precipitation is around 534 mm, and 90% precipitation occurs in May to September [49]. The primary soil type in this area is Mollisols (also called Black soils), accounting for 64.2% of the total area. The structure of the soil profile is A (the topsoil which is porous and fertile), AB (the transition layer), B (the illuvial horizon) and C (the parent material which has a poor structure and low fertility). The topsoil of black soil is

dominated by large pores and exhibits a high water-holding capacity. The under layer has a heavy clay texture, poor structure, and low water permeability. These properties allow runoff to form readily in the surface layer, not only resulting in the loss of soil nutrients, but also in damage to the soil's physical properties, and weaken the soil infiltration capacity [49,53]. Soybean is one of the most important summer crops in this area, with a water requirement of 370 to 540 mm [54], which is sown in early May and reaped in late September.

### 2.2. Pot Experiments

To measure the effects of soil water on crop growth processes for different eroded soils, the potted water-control experiments for soybean were conducted for four years (from June to September 2013, and May to September 2014, 2015, and 2016).

#### 2.2.1. Soil Selection and Pots Collection

According to the typical soil profile of A-AB-B-C in the black soil region, the structure of the soil profiles were used for soil erosion degree estimation. Three undisturbed eroded soils (lightly, moderately, and severely) from the original field of the typical black soil were selected by soil profile depths. This was determined by sampling, using a soil auger. The characters of the three eroded soils were listed in Table 1. Soil physicochemical properties for the top 30 cm layer for each of the eroded soils are listed in Table 2.

**Table 1.** Classification of soil erosion.

| Erosion Degree | Indication | Sample Location |
|----------------|------------|-----------------|
| Lightly | 30 cm depth of horizon A remaining. | Contour tillage in the 2° slope |
| Moderately | 20 cm depth of horizon A remaining. | Conventional tillage in the 2.4° slope |
| Severely | 0 cm depth of horizon A remaining. | Conventional tillage in the 3.1° slope |

A, mineral horizon.

**Table 2.** The physicochemical property of three eroded soils.

| Types | FC (%) | WP (%) | Clay (%) | Silt (%) | Sand (%) | BD (g cm$^{-3}$) | SOM (%) | AN (mg kg$^{-1}$) | PH | AP (mg kg$^{-1}$) | AK (mg kg$^{-1}$) |
|-------|--------|--------|----------|----------|----------|------------------|---------|-------------------|----|-------------------|-------------------|
| L | 38.8 | 15.2 | 34.88 | 34.30 | 30.82 | 1.23 | 5.85 | 241 | 5.69 | 18.7 | 41.36 |
| M | 25.5 | 10.1 | 31.71 | 24.95 | 43.34 | 1.47 | 4.56 | 157 | 5.96 | 33.22 | 65.33 |
| S | 17.3 | 6.7 | 16.52 | 15.35 | 68.13 | 1.67 | 1.96 | 157 | 5.83 | 41.46 | 49.35 |

L, lightly eroded soil; M, moderately eroded soil; S, severely eroded soil; A, mineral horizon; FC, field capacity; WP, wilting point; BD, bulk density; SOC, soil organic matter; AN, available nitrogen; AP, available phosphorus; AK, available potassium.

Eroded soils for the experiment were collected with a soil corer consisting of a PVC pipe 35 cm tall and 16 cm diameter with a 16.5 cm diameter cutting ring attached. An electric drill with high-frequency oscillations drove the samples into the ground. A total of 180 potted plants (four moisture conditions × three erosion intensities × five growth stages × three replicates) with undisturbed soil were randomly placed under three canopies (length: 26 m; width: 3 m) for each year. The shelters were opened in the absence of rain to ensure sunshine and ventilation, and closed before rain to ensure that they were not affected by rainfall.

#### 2.2.2. Soil Water Selection and Control for Each Pot

To determine the effects of different water conditions on the growth of soybean, four water control conditions were selected, which were 100%, 80%, 60%, and 40% of the field water holding capacity (FC) of the three types of eroded soils (FC of <40% is not suitable for crop growth [55]. Soil moisture in all of the pots was maintained at 80%–75% FC after sowing to ensure seeding emerge, and when

two leaves sprouted in all the pots, according to the test requirements, we began to control the water content every other day.

The weighting method used to control soil water for each pot was based on the following equation:

$$DT_i = DW_i - W_0 + S + P, \tag{1}$$

where i is one of the four soil water content levels, $DT_i$ is the total weight of a pot filled with soil at the ith designated soil water content level (g), $DW_i$ is the soil water weight for the ith designated soil water content level based on the FC for each eroded soil (g), $W_0$ is the initial soil water weight when the soil was sampled for each pot (g), S is the dry soil weight for each pot (g), and P is the empty pot weight measured before the pot was filled with soil (g). The estimates of the designated soil water weight ($DW_i$) for the three eroded soils in the dry soil weight for each pot (S) was multiplied by the soil water content relative to FC (100%, 80%, 60%, or 40%). Both $W_0$ and S are calculated using the weighted average of the measured soil water content for the 0–10 cm, 10–20 cm, and 20–30 cm layers during the soil sampling for each pot. If the pot weight is less than the calculated $DT_i$ for a particular pot, water was added until the pot weight was equal to $DT_i$.

### 2.2.3. Crop Growth Indexes Measurement

The soybean growing period was divided into five stages—branching, flowering, pod formation, seed filling, and maturity—to analyze crop growth processes. When 85%–90% of the soybeans in pots had the characteristics for a stage, 36 pots of the three eroded soils and four soil water levels with three replications were broken. The branching stage occurred 33–46 days after sowing, followed by the flowering stage at 51–64 days after sowing, the pod formation stage at 70–82 days, the seed filling stage at 88–102 days, and the maturity stage at 100–126 days after sowing.

Since the focus on the study was WUE, we measured the indexes of evapotranspiration (ET), biomass (B) and yield (Y) during the entire growth of the crop. After the first euphylla appeared, we began to weigh each pot every other day to calculate evapotranspiration. Aboveground biomass (B) was measured at each growth stage. Two soybean plants were cut in the pot with shear. The aboveground biomass was dried by an oven at 80 °C and the weight was written down. Yield (Y) was measured at maturity. The naturally dried beans in each pot were weighed.

### 2.3. Soil Physicochemical Properties

Soil samples were collected in layers of 0–15 cm and 15–30 cm, separately at each sampling point. The mechanical composition, organic matter, bulk density, wilting points, and available nitrogen, phosphorus and potassium were measured for each sample. (1) Soil samples taken from each depth were mixed together. The natural air-dried samples were sieved through 2 mm and pipette method was used to determine mechanical composition. And the natural air-dried samples were sieved through 0.149 mm and used the potassium dichromate volumetric method to determine soil organic matter. (2) The undisturbed soil was collected using 100 $cm^3$ cutting rings three repetitions of each depth. A total of 18 cutting rings were collected to determine soil saturated water content and bulk density. (3) Wilting points were measured with pressure membrane apparatus. The undisturbed soil was collected using a rigid PVC ring with a diameter of 4 cm and height of 1 cm with six repetitions each, and 36 soil samples were collected. (4) Available nitrogen, phosphorus and potassium were measured with alkaline hydrolysis-diffusion method, sodium bicarbonate extraction technology and ammonium acetate extraction flame photometer, respectively.

### 2.4. Data Analyses

Harvest index (HI), as a link between yield and biomass, was calculated, which influences the assimilation ratio assigned to the grain [56]. The related formula is as follows

$$HI = Y/B \tag{2}$$

$$B = \left[ b \times 100/\pi R^2 \right] \times 0.36 \tag{3}$$

$$Y = \left[ y \times 100/\pi R^2 \right] \times 0.36 \tag{4}$$

where HI is the harvest index, Y is the crop economic yield (t ha$^{-1}$), and B is the above-ground biomass (t ha$^{-1}$), b and y are the weights of dry biomass and yield (g), respectively.

Water use efficiency (WUE) is often considered an important determinant of yield under stress and even as a component of crop drought resistant [57]. It can be expressed as follows

$$\text{Based on dry biomass mass}: \ WUE_B = B/ET \tag{5}$$

$$\text{Based on economic yield}: \ WUE_Y = Y/ET \tag{6}$$

$$ET = \frac{\left[ (DT_i - PW_i) \times 10/\pi R^2 \right]}{d} \times 0.36 \tag{7}$$

where, ET is the evapotranspiration (mm), $WUE_B$ is the WUE based on the biomass (kg ha$^{-1}$ mm$^{-1}$), and $WUE_Y$ denotes the yield-based WUE (kg ha$^{-1}$ mm$^{-1}$), i is one of the four soil water content levels, $DT_i$ is the total weight of a pot filled with soil at the ith designated soil water content level (g), $PW_i$ is the weight of a pot that record every other day (g), R is the radius of a pot with the value of 8 (cm), d is the interval days for measured with the value of 2 (day).

The crop yield response coefficient (Ky) is a comprehensive description of the relationship between crop yield and water stress [58,59]. Lovelli et al. [60] established the relationship between biomass and water consumption according to the response relationship of the crop growth to the water supply in different growth periods. The equations are as follows

$$K_y = \frac{1 - Y_a/Y_m}{1 - ET_a/ET_m} \tag{8}$$

$$K_b = \frac{1 - B_a/B_m}{1 - ET_a/ET_m} \tag{9}$$

where $Y_a$ and $B_a$ are the actual crop yield (t ha$^{-1}$) and crop biomass (t ha$^{-1}$), respectively; $Y_m$ and $B_m$ are the maximum yield (t ha$^{-1}$) and biomass (t ha$^{-1}$), respectively; $ET_a$ is the actual evapotranspiration (mm), $ET_m$ is the maximum evapotranspiration (mm), and $K_y$ and $K_b$ is the crop yield (or biomass) to water supply response coefficient. $K_y$ (or $K_b$) > 1 indicates that yield (biomass) is very sensitive to water consumption reduction; $K_y$ (or $K_b$) < 1 indicates that the yield (biomass) is not sensitive to the reduction in water consumption.

To quantitatively reflect the effect of water stress on WUE, the coefficient of stress (*SF*) is defined as the ratio of WUE under actual water conditions to the optimal water supply as follows [61]

$$SF = \frac{WUE_a}{WUE_m} \tag{10}$$

where $WUE_a$ is WUE under the actual water conditions, $WUE_m$ is WUE under the optimal water supply, SF is the coefficient of stress of WUE, and its value is between 0 and 1. SF = 1 indicates that the crop growth is not affected by the water stress; SF < 1 indicates that the crop growth is influenced by water stress; SF = 0 indicates that the water stress causes the crop to die.

By setting the regression equation, quantitative analysis reveals the relationships between erosion–water use–productivity.

## 3. Results

### 3.1. B, ET, Y, and HI under the Optimal Soil Moisture Conditions

The biomass (B), evapotranspiration (ET), and yield (Y) of the values exhibited similar differences, for three eroded soils, but no significant difference was observed in HI (Figure 1). For B and Y, the optimal water content was 80% FC for lightly eroded soil (L) and 100% FC for both moderately eroded soil (M) and severely eroded soil (S). For ET, the optimal water content was 100% FC for the three eroded soils. Under the conditions of adequate water supply, there was no significant difference between L and M, but the values were significantly different from the S ($p < 0.05$). With an increase in the soil water stress, the values for L were significantly different from those for M and S, and no significant difference was observed between M and S (Figure 1).

Under optimal water supply conditions, a comparison of the three eroded soils revealed that the B of L was the highest, at 5.89 t ha$^{-1}$, which was 1.2% and 81.3% higher than M and S, respectively. The Y value of M was the highest, at 3.12 t ha$^{-1}$, which was 4.6% and 85.5% higher than L and S, respectively. The ET in L was the best with the value of 354 mm, 1.8% higher than that of M and 30.6% higher than that of S. HI ranged from 0.50 to 0.55, with an average of 0.51 (Figure 1).

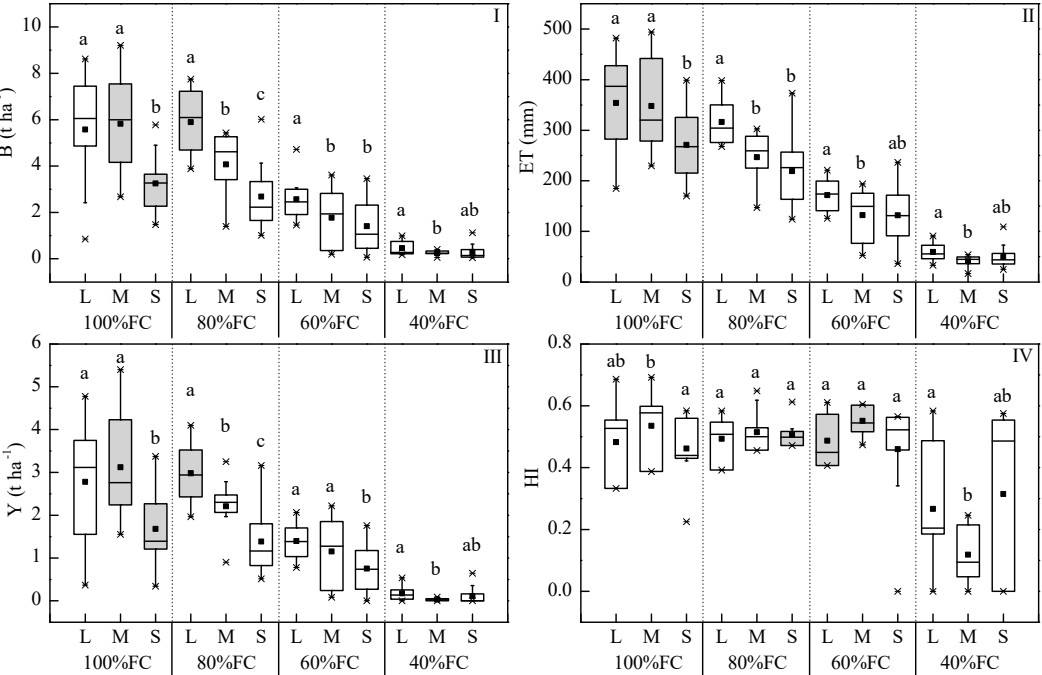

**Figure 1.** Effects of soil erosion on B (**I**), ET (**II**), Y (**III**) and HI (**IV**) under four different water conditions for the entire growth period. B, biomass. ET, evapotranspiration. Y, yield. HI, harvest index. L, lightly eroded soil. M, moderately eroded soil. S, severely eroded soil. Different letters indicate significant differences at $p = 0.05$ (paired-samples *t*-tests) for different eroded soils. The gray boxes denote the indicators for optimal moisture conditions.

The findings for the different growth stages were similar to those for the entire growth period (Figure 2). No significant difference in B and ET were observed between L and M, but these values were significantly different from those of S ($p < 0.05$) under optimal water supply conditions. The daily water consumption was the largest in the pod formation stage. The average daily ET in this stage of L was 5.63 mm, which was 23.7% and 55.5% higher than in M and S, respectively, and the B value of L was 5.35 t ha$^{-1}$, which was 19.2% and 89.2% higher than M and S, respectively.

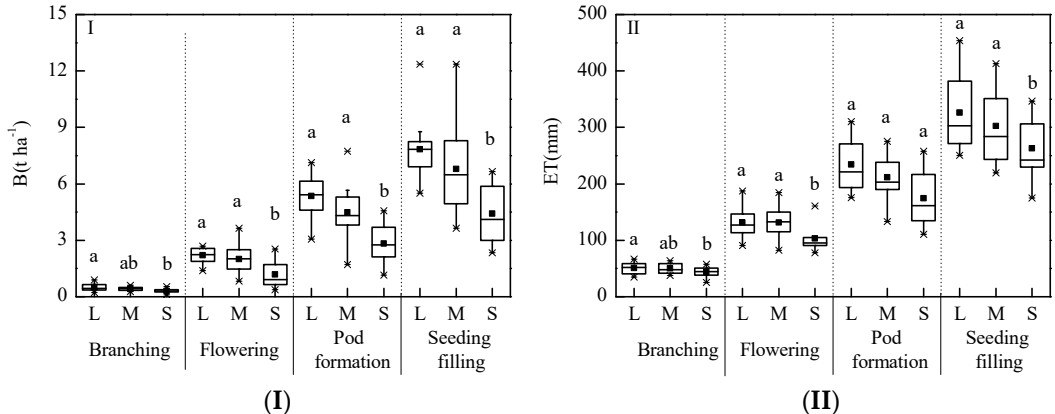

**Figure 2.** Effects of soil erosion on B (**I**) and ET (**II**) under optimal water conditions in different growth stages. B, biomass. ET, evapotranspiration. Different letters indicate significant differences at *p* = 0.05 (paired-samples *t*-tests) for different eroded soils.

### 3.2. Yield Response to Water Stress

Although there were significant differences between the three eroded soils for $WUE_Y$, when the relative water contents were considered, the response functions of the soybean yield to water stress were linear and not significantly different for the three eroded soils in the tests. Therefore, the Y and ET values of the three eroded soils were fitted to obtain the yield response coefficient (Ky) of 1.12 for the entire growth period (Figure 3I), which implies that Y was sensitive to water stress.

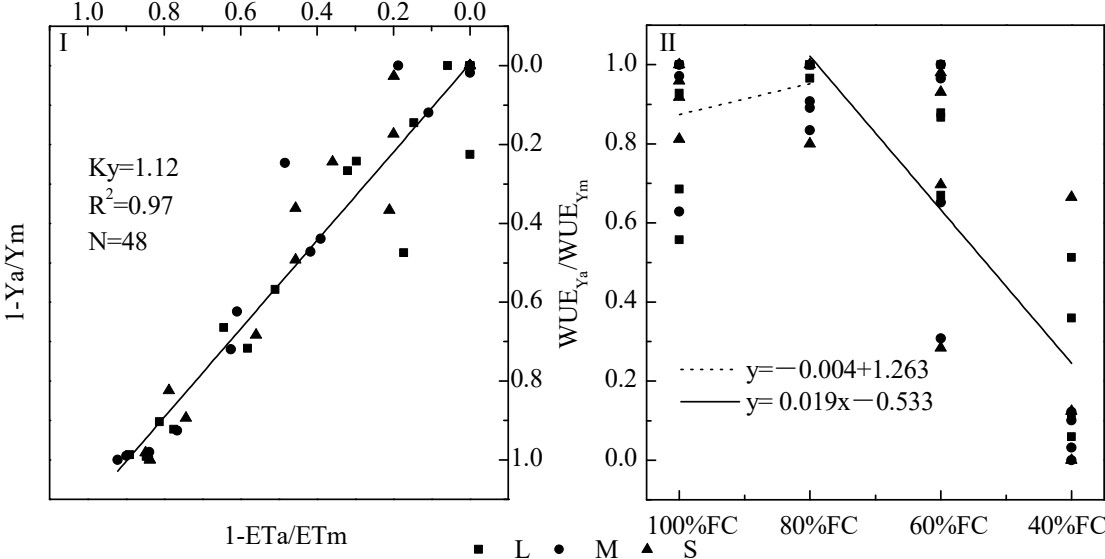

**Figure 3.** Yield response to water stress curves (**I**), and stress factor of water use efficiency ($WUE_Y$) stress coefficient (SF) (**II**) for the entire growth period.

No significant differences were observed in the crop biomass to water supply response coefficient (Kb) in the different growth stages. The fitting results showed that the Kb value for the branching period was 0.90 and that for the flowering period was 1.10, and the values for both the pod formation and the seed filling periods were 1.11 (Figure 4). The results showed that the water stress was not obvious in the branching stage, but B is sensitive to water stress in the flowering, pod filling and seeding filling stage.

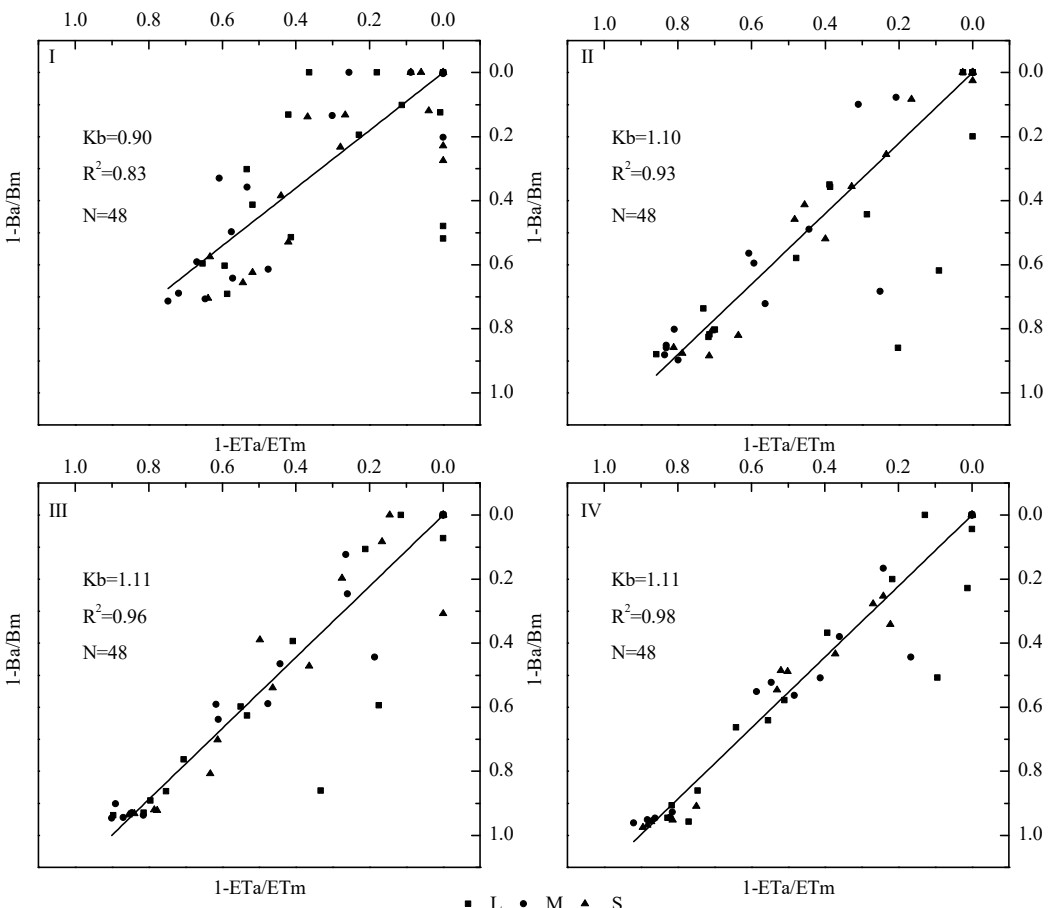

**Figure 4.** Yield response to water stress curves for different growth stages. (**I**), Branching; (**II**), flowering; (**III**), pod filling; (**IV**), seeding filling.

*3.3. WUE-SF under Water Stresses for Different Eroded Soils*

3.3.1. Water Use Efficiency under the Optimal Soil Moisture Conditions

The $WUE_Y$ and $WUE_B$ of soybean in soils with different degrees of erosion exhibited certain discrepancies (Figure 5). The values of $WUE_Y$ and $WUE_B$ for the three eroded soils were best at 80% FC. For $WUE_Y$, no significant difference was observed between L and M, but both of these values were significantly different from the value of S ($p < 0.05$). However, for $WUE_B$, there were significant differences for the three eroded soils under optimal water supply conditions. The values of $WUE_Y$ and $WUE_B$ for L were 9.36 kg ha$^{-1}$ mm$^{-1}$ and 17.48 kg ha$^{-1}$ mm$^{-1}$, which were 5.1% and 11.7% higher than that of M, and 51.9% and 35.2% higher than that of S, respectively. In the waterlogged state, the $WUE_Y$ and $WUE_B$ of M were the best, while, in the water stress states, the $WUE_Y$ and $WUE_B$ of L were the best.

In different growth stages, the optimal water supply conditions of $WUE_B$ for the three eroded soils were 40% FC in the branching stage, and 80% FC in the flowering, pod filling and seeding filling stage. During the first three stages, no significant difference was observed between the values of L and M, but both of them were significantly different from the value of S (p < 0.05). In the seed filling stage, significant differences were observed between the three eroded soils (p < 0.05), and the values of $WUE_B$ were the best in all stages. The $WUE_B$ value of L was 25.34 kg ha$^{-1}$ mm$^{-1}$, which was 1.2 and 1.5 times higher than that of L and S, respectively (Figure 6).

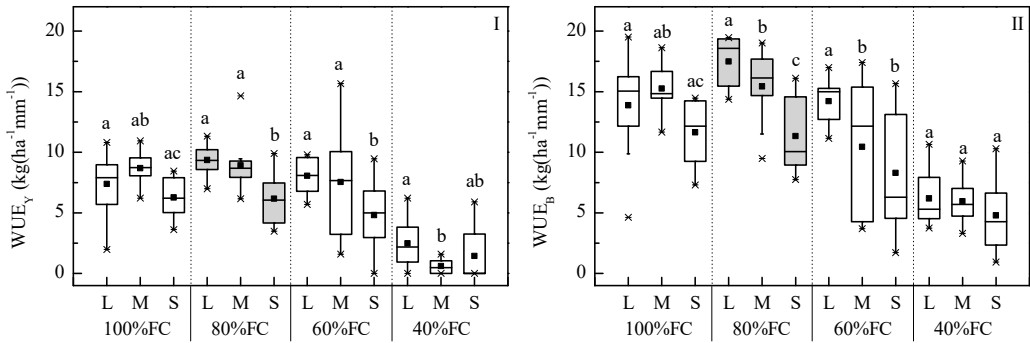

**Figure 5.** Effects of soil erosion on WUE under optimal water conditions in entire growth period. (**I**), WUE$_Y$, WUE based on yield. (**II**), WUE$_B$, WUE based on biomass. Different letters indicate significant differences at *p* = 0.05 (paired-samples *t*-tests) for different eroded soils.

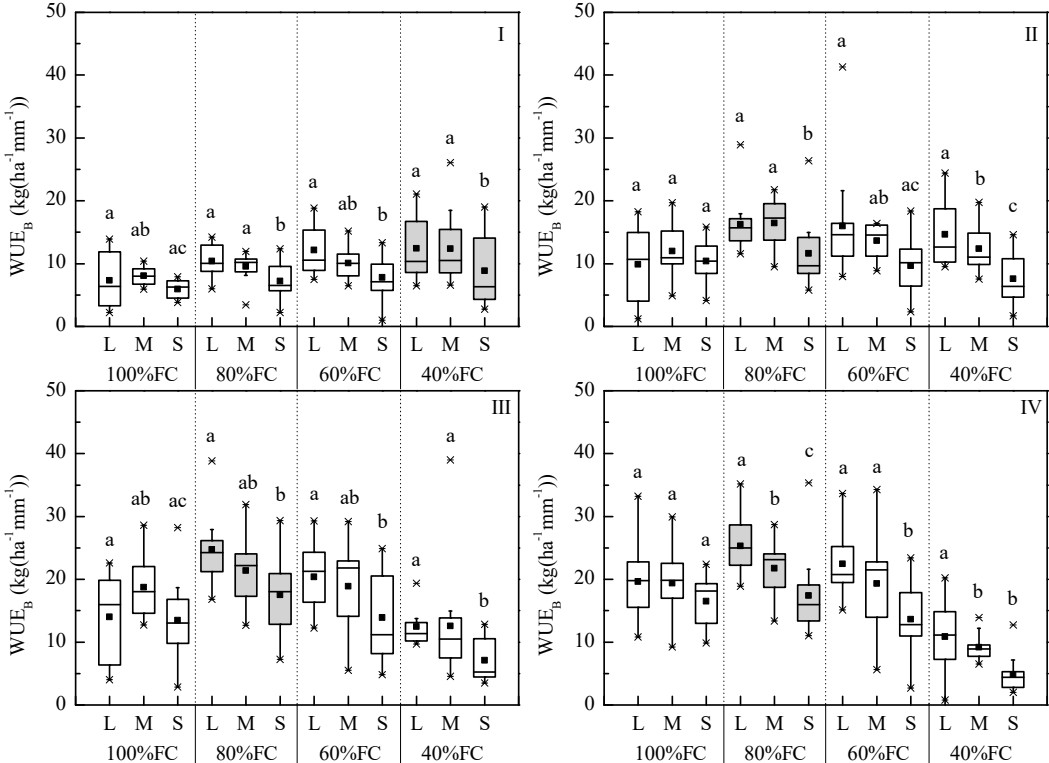

**Figure 6.** Effects of soil erosion on WUE under optimal water conditions in different growth stages. (**I**), Branching; (**II**), flowering; (**III**), pod filling; (**IV**), seeding filling. Different letters indicate significant differences at *p* = 0.05 (paired-samples *t*-tests) for different eroded soils.

### 3.3.2. WUE-SF Response to Water Stress

In entire growth period, the SF values of the three eroded soils were not significantly different. The fitting results indicate (Figure 3II) that under waterlogging stress conditions, SF was 0.004, which implies that when the soil moisture increased by 10%, WUE$_Y$ decreased by 4%. In the case of water shortage, SF was 0.019, which implies that when the soil moisture decreased by 10%, WUE$_Y$ decreased by 19%, which was 4.8 times that under waterlogging stress.

In different growth stages, the results showed that the water stress was not obvious in the branching stage. The effect of the waterlogging stress on soybean was higher than that of the water shortage stress during the flowering period. Moreover, under the condition of waterlogging stress, the soil water content increased by 10% and WUE$_B$ decreased by 10%, and under the condition of water shortage stress, the soil water content decreased by 10% and WUE$_B$ decreased by 6%, which implies

that it was 0.6 times more powerful than the waterlogging stress of WUE$_B$. In the pod formation stage, the response of soybean to the waterlogging stress was a little smaller than the water shortage stress, and the soil water decreased (or increased) by 10% and WUE$_B$ decreased by 12% and 11%, respectively. The effect of the water shortage stress on the soybean was higher than that of the waterlogging stress in the seed filling stage. Under the water shortage stress, the soil water content decreased by 10% and WUE$_B$ decreased by 15%, which was 2.5 times more powerful than the waterlogging stress (Figure 7).

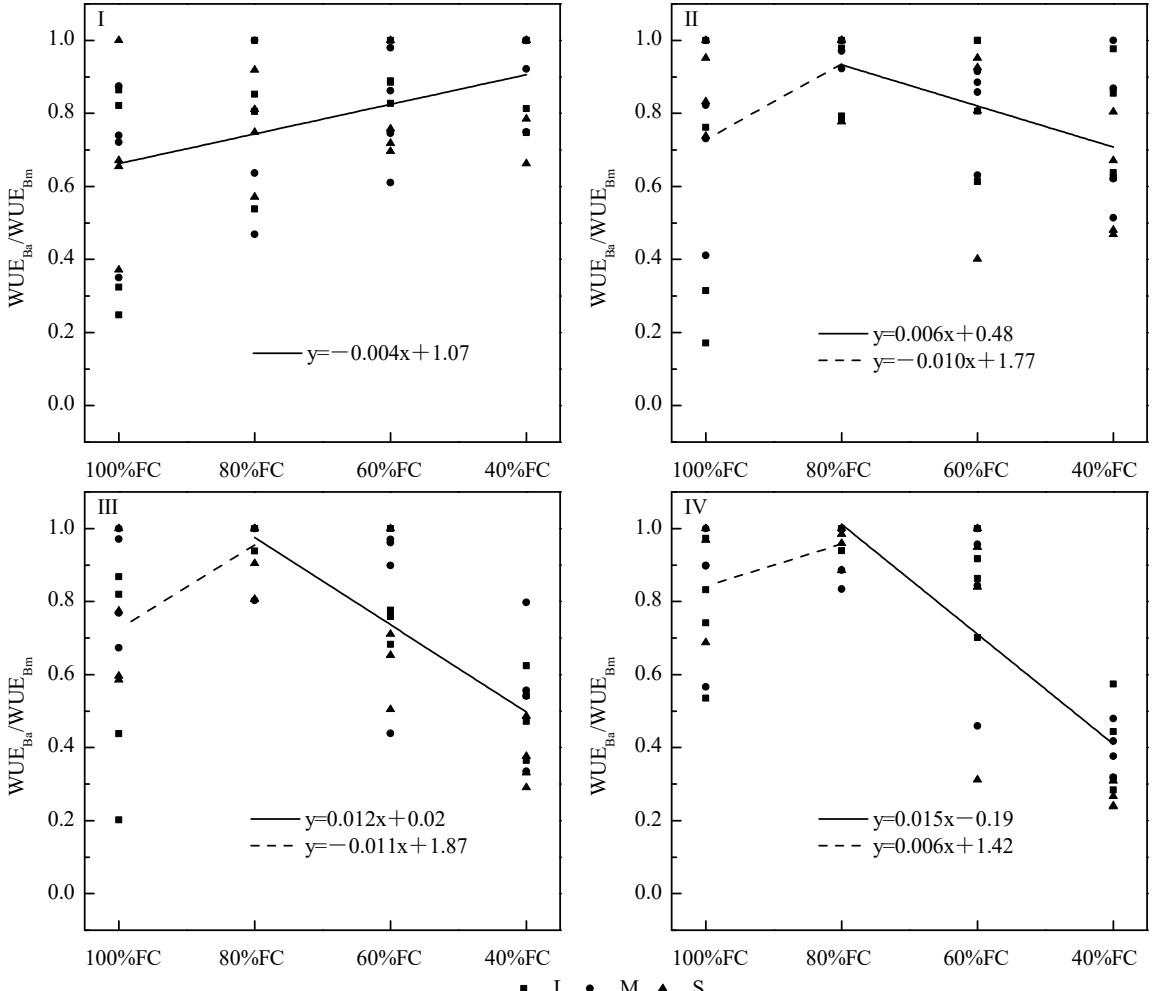

**Figure 7.** Stress factor of WUEB (SF) for different growth stages. (**I**), Branching; (**II**), flowering; (**III**), pod filling; (**IV**), seeding filling.

## 4. Discussion

### 4.1. Optimal Moisture Conditions for Different Eroded Soils in Y and B

In this study, the optimal water content was 80% FC for L and 100% FC for both M and S (Figure 1). This finding might be caused by available soil moisture and water holding capacity (Table 2). Frye et al. [24] and Battiston et al. [26] showed that soil erosion affects the soil's organic matter transport and reduces the soil's water holding capacity. After soil erosion in the 0–30 cm soil layer and a significant decrease in the water holding capacity, the soil water capacity was 4%–6% lower than that of soil without obvious erosion. Wendt et al. [25] and Murphree et al. [27] by comparing the soil infiltration after erosion, found that the soil infiltration was reduced by the middle-degree erosion, and, for crop growth, the effective use of water was reduced by approximately 7%–44%. Andraski and Lowery [28] compared the effective moisture of the L, M, and S soils and found that the effective moisture of M and

S was lower than that of L by 7% and 14%, respectively. In this study, the L with field capacity was 38.8%, its wilting point was 13.9%, and soil available water capacity (AWC) was 23.6%. After erosion, the value of AWC decreased to 15.4% and 10.6% for M and S, respectively. For M and S, the lower of the effective water content of the soil is, the more the soil needs to maintain the maximum effective water for crop growth. Therefore, as the soil moisture of M and S was less than 100%FC, the soybean growth stress was observed.

Some studies of crop models, such as EPIC [62], showed that waterlogging stress was 103% FC, and water shortage stress was 52% FC, which was 28% lower than the critical soil water capacity of L for water shortage stress and 48% lower than M and S, respectively. The critical soil water capacity of L was 12% and 22% higher than the critical value of stomatal conductance (68% FC) and canopy attenuation (58% FC) in the AquaCrop model [59,63], and, for M and S, was 32% and 42% higher than stomatal conductance (68% FC) and canopy attenuation (58% FC), respectively, in the AquaCrop model. The critical water content of waterlogging stress in L was about 20% lower than EPICdid were not reflected the difference in water stress among different eroded soils. The results provide important parameters for the growth of simulated crops.

Yield was best in M under optimal moisture which might be caused by soil texture changes after erosion (Table 2). The L texture was clay loam, according to the soil texture classifications defined by the IS, with a sand content of 30.82%, a silt content of 34.3% and a clay content of 34.88%. For M and S, the contents of silt were 24.95% and 15.35%, and the clay contents were 31.71% and 16.52%, respectively, however, the sand contents for both were 43.34% and 68.13%, respectively. Correspondingly, the bulk densities increased from 1.23 to 1.67 g cm$^{-3}$ and the organic matter content decreased from 5.85% to 1.96% from L to S, respectively. Notably, the M still had an organic matter content as high as 4.56% which with no nutrient limitations. But, for L, the poor drainage resulting from the high clay content and low bulk density could cause deficient soil aeration and thereby limit root growth, resulting in reduced production. Many studies have confirmed that soil with lower bulk density and higher clay content could lead to the restriction of root growth and further reduce the yield [64–70].

### 4.2. Yield and Biomass Response to Water Stress

In this study, soil erosion had no significant effect on the response of soybean biomass and yield to water stress, and the corresponding coefficient of yield was 1.10–1.12. The result of this study was different from that of Doorenbos et al. [58], who argued that during the entire growth period, Ky was 0.85 and was not sensitive to a water deficit. However, this result was similar to the studies of Rosadi et al. [71] who reported that Ky was 1.05, and Stegman et al. [72] estimated Ky to be 1.26. The findings of both these studies are more or less in agreement with those of the current study, suggesting that soybean yield is more sensitive to water stress.

In the different growth stages, the response of biomass to water stress was more than 1.0, which reflects that reproductive growth period is more sensitive to water stress [73,74]. Eck et al. [75] reported that soybean seed yield is least sensitive to water deficits during the vegetative stage, more sensitive during flowering and pod set, and most sensitive during pod fill. Wen and Zhao [76] used the Jensen model to simulate spring soybean; this model showed that the yield response to the water stress of soybean was different for different growth stages. The sensitivity of soybean before the flowering stage was the lowest, however, in the flowering to pod formation period, it was the largest. This result was similar to my study.

### 4.3. WUE-SF under Water Stresses for Different Eroded Soils

In this study, the WUE for L was the best. WUE was affected by water stress and soil erosion, and WUE decreased with an increase in water stress [77]. Soil fertility and organic matter will be changed by soil erosion [11–14]. When the soil organic matter content was decreased significantly, such as from 5.85% to 1.96% in this study, the available N in the soil decreased from 241 mg kg$^{-1}$ to

157 mg$^{-1}$ (Table 2), and the nutrient content decreased. The resulting structural damage significantly decreased WUE.

It is important to recognize these critical growth stages of crop water requirement. Some studies for millet reported that water stress at the ear emergence stage caused the greatest reduction, and erosion at vegetative and seed filling stage had no significantly effect on WUE [78–81]. For soybean, Baghel et al. [82] showed that water stress at the flowering stage severely decreased all of the above parameters in soybean. Jaybhay et al. [83] reported that irrigation to the soybean crop at flower initiation and seed filling stages helps to obtain the optimum WUE. Currently, many studies only focused on the critical growth stages, not quantitative analysis on the stress factors of WUE–SF for soybean. Although the stress was not affected by the soil properties in this research, it showed obvious differences in the different growth periods. Before the flowering period, the influence of the waterlogging stress was greater than that of the water deficit stress, and when the soil moisture increased by 10%, WUE$_B$ decreased by approximately 10%. The effect of the water deficit stress was more significant than that of the waterlogging stress after the flowering period. With the soil water content decreased by 10%, WUE$_B$ decreased by approximately 10% in the pod formation period. Moreover, the soil moisture decreased by 10% and WUE$_B$ decreased by 20% in the seed filling period. Therefore, the seed filling period is a critical growth stage for soybean in northeast China.

## 5. Conclusion

In this study, three erosion degrees, (1) lightly, (2) moderately and (3) severely eroded black undisturbed soils, were collected to study the relationships between erosion–water use–productivity. The following results were achieved:

(1) For B and Y, the optimal water content was 80% FC, for L, and 100% FC, for both M and S. Yield was best in M, with a value of 3.12 t ha$^{-1}$, which was 4.6% and 85.5% higher than L and S, respectively. For ET, the optimal water content was 100% FC for three eroded soils. Under the conditions of adequate water supply, there was no significant change in Land M, but values were significantly different for the S ($p < 0.05$);

(2) The values of WUEY and WUEB for the three eroded soils were the best at 80% FC. The value of WUEY and WUEB for L was 9.36 kg ha$^{-1}$ mm$^{-1}$ and 17.48 kg ha$^{-1}$ mm$^{-1}$, which was 5.1% and 11.7% higher than that of M, and 51.9% and 35.2% higher than that of S, respectively. In different growth stages, the optimal water supply conditions of WUEB for three eroded soils were 40% FC in the branching stage, and 80% FC in other stages;

(3) Y and B were sensitive to water stress except in the branching stage. There was no significant difference among the three eroded soils in the functions of the soybean yield after water stress;

(4) The SF values of the three eroded soils were not significantly different. In the entire growth period, the SF value was 0.019 in the case of water shortage, which implies that, when the soil moisture decreased by 10%, WUEY decreased by 19%, which was 4.8 times that under waterlogging stress. In the flowering and pod formation stage, the SF reached the maximum under waterlogging stress, the soil water content decreased by 10% and WUEB decreased by 10%. In the seed filling stage, the SF reached the maximum under the water shortage stress, the soil water content decreased by 10% and WUEB decreased by 15%, which was 2.5 times more powerful than the waterlogging stress.

**Author Contributions:** Data curation, S.L., Y.X. (Yun Xie), G.L., J.W., H.L. and J.Z.; Formal analysis, S.L. and Y.X. (Yun Xie); Funding acquisition, Y.X. (Yun Xie); Methodology, S.L. and Y.X. (Yun Xie); Supervision, Y.X. (Yun Xie); Writing – original draft, S.L.; Writing – review & editing, S.L., Y.X. (Yun Xie), H.L. and Y.X. (Yan Xin). All authors have read and agreed to the published version of the manuscript.

**Funding:** This work was supported by the National Key R&D Plan (grant number 2017YFE0118100), and the National Key R&D Program (No. 2018YFC0507000).

**Conflicts of Interest:** The authors declare no conflict of interest.

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
