# Peer review of "Water Use Efficiency of Soybean under Water Stress in Different Eroded Soils"

_water, doi:10.3390/w12020373_

Round 1

Reviewer 1 Report

Interesting paper on an important issue. However some improvemnets are needed to become more undrstandable and clear. See comments marked on the attached manuscript. I suggest the paper's acceptance after making the improvments marked on the manuscript.

Author Response

Response to Reviewer1 Comments

Interesting paper on an important issue. However some improvements are needed to become more understandable and clear. See comments marked on the attached manuscript. I suggest the paper's acceptance after making the improvements marked on the manuscript.

Point 1: To which horizons they belong, lightly, moderately or severely eroded soil? Explain.

Response 1: Thanks for your suggestion. We did not describe this clearly. We have added sentences to explain this: “According to the typical soil profile of A-AB-B-C in the black soil region, the depth of A and B horizons to the parent material horizons were used for soil erosion degree estimation. Three undisturbed eroded soils (lightly, moderately, and severely) from the original field of the typical black soil were selected by soil profile depths.” (Line 102 to 109 on page 3)

Table 1. Classification of soil erosion.

Erosion degree

Indication

Sample location

Lightly

30 cm depth of horizon A remaining.

Contour tillage in the 2° slope

Moderately

20 cm depth of horizon A remaining.

Conventional tillage in the 2.4° slope

Severely

0 cm depth of horizon A remaining.

Conventional tillage in the 3.1° slope

A, mineral horizon

Point 2: You should add some more soil proerties such as pH, calcium carbonate content, available phosphorus and potassium. And explain the meaning of the symbols (A, FC and WP).

Response 2: We have added some more soil properties in Table 2, and explained the meaning of the symbols: “L, lightly eroded soil; M, moderately eroded soil; S, severely eroded soil; A, mineral horizon; FC, field capacity; WP, wilting point; BD, bulk density; SOC, soil organic carbon; AN, available nitrogen; AP, available phosphorus; AK, available potassium.” (Line 110 to 120 on page 3)

Table 2. The physicochemical property of three eroded soils

Types

FC

(%)

WP

(%)

Clay

(%)

Silt

(%)

Sand

(%)

BD

(g cm-3)

SOM

(%)

AN

(mg kg-1)

PH

AP

(mg kg-1)

AK

(mg kg-1)

L

38.8

15.2

34.88

34.30

30.82

1.23

5.85

241

5.69

18.7

41.36

M

25.5

10.1

31.71

24.95

43.34

1.47

4.56

157

5.96

33.22

65.33

S

17.3

6.7

16.52

15.35

68.13

1.67

1.96

157

5.83

41.46

49.35

L, lightly eroded soil; M, moderately eroded soil; S, severely eroded soil; A, mineral horizon; FC, field capacity; WP, wilting point; BD, bulk density; SOC, soil organic carbon; AN, available nitrogen; AP, available phosphorus; AK, available potassium.

Point 3: Give the pots' dimensions.

Response 3: We have added sentences to explain this: “Eroded soils for the experiment were collected with a soil corer consisting of a PVC pipe 35 cm tall and 16 cm diameter with a 16.5 cm diameter cutting ring attached.” (Line 114 to 115 on page 3)

Point 4: Describe clearly the experimental design, treatments, replications.

Response 4: Thanks for your suggestions. We have rewritten the section of 2.2 pot experiments in the manuscript. In that, the experimental design, treatments, and replications were clear.

“2.2. Pot experiments

To measure the effects of soil water on crop growth processes for different eroded soils, the potted water-control experiments for soybean were conducted for four years (from June to September 2013, and May to September 2014, 2015, and 2016).

2.2.1. Soil selection and pots collection

According to the typical soil profile of A-AB-B-C in the black soil region, the depth of A and B horizons to the parent material horizons were used for soil erosion degree estimation. Three undisturbed eroded soils (lightly, moderately, and severely) from the original field of the typical black soil were selected by soil profile depths. This was determined by sampling, using a soil auger. The characters of the three eroded soils were listed in Table 1. Soil physicochemical property for top 30cm layer for each of the eroded soils were listed in Table 2.

Table 1. Classification of soil erosion.

Erosion degree

Indication

Sample location

Lightly

30 cm depth of horizon A remaining.

Contour tillage in the 2° slope

Moderately

20 cm depth of horizon A remaining.

Conventional tillage in the 2.4° slope

Severely

0 cm depth of horizon A remaining.

Conventional tillage in the 3.1° slope

A, mineral horizon

Table 2. The physicochemical property of three eroded soils

Types

FC

(%)

WP

(%)

Clay

(%)

Silt

(%)

Sand

(%)

BD

(g cm-3)

SOM

(%)

AN

(mg kg-1)

PH

AP

(mg kg-1)

AK

(mg kg-1)

L

38.8

15.2

34.88

34.30

30.82

1.23

5.85

241

5.69

18.7

41.36

M

25.5

10.1

31.71

24.95

43.34

1.47

4.56

157

5.96

33.22

65.33

S

17.3

6.7

16.52

15.35

68.13

1.67

1.96

157

5.83

41.46

49.35

L, lightly eroded soil; M, moderately eroded soil; S, severely eroded soil; A, mineral horizon; FC, field capacity; WP, wilting point; BD, bulk density; SOC, soil organic matter; AN, available nitrogen; AP, available phosphorus; AK, available potassium.

Eroded soils for the experiment were collected with a soil corer consisting of a PVC pipe 35 cm tall and 16 cm diameter with a 16.5 cm diameter cutting ring attached. An electric drill with high-frequency oscillations drove the samples into the ground. A total of 180 potted plants (4 moisture conditions × 3 erosion intensities × 5 growth stages× 3 replicates) with undisturbed soil were randomly placed under three canopies (length: 26 m; width: 3 m) for each year. The shelters were opened in the absence of rain to ensure sunshine and ventilation, and closed before rain to ensure that they were not affected by rainfall.

2.2.2. Soil water selection and control for each pot

To determine the effects of different water conditions on the growth of soybean, four water control conditions were selected, which were 100%, 80%, 60%, and 40% of the field water holding capacity (FC) of the three types of eroded soils (FC of <40% is not suitable for crop growth [56]. Soil moisture in all of the pots was maintained at 80%–75% FC after sowing to ensure seeding emerge, and when two leaves sprouted in all the pots, according to the test requirements, we began to control the water content every other day.

The weighting method used to control soil water for each pot was based on the following equation:

                           ,                                (1)

where i is one of the four soil water content levels,  is the total weight of a pot filled with soil at the ith designated soil water content level (g),  is the soil water weight for the ith designated soil water content level based on the FC for each eroded soil (g),  is the initial soil water weight when the soil was sampled for each pot (g),  is the dry soil weight for each pot (g), and  is the empty pot weight measured before the pot was filled with soil (g). The estimates of the designated soil water weight () for the three eroded soils in the dry soil weight for each pot () was multiplied by the soil water content relative to FC (100%, 80%, 60%, or 40%). Both  and  are calculated using the weighted average of the measured soil water content for the 0–10 cm, 10–20 cm, and 20–30 cm layers during the soil sampling for each pot. If the pot weight is less than the calculated  for a particular pot, water was added until the pot weight was equal to .

2.2.3. Crop growth indexes measurement

The soybean growing period was divided into five stages: branching, flowering, pod formation, seed filling, and maturity to analyze crop growth processes. When 85–90% of the soybeans in pots had the characteristics for a stage, 36 pots of the three eroded soils and four soil water levels with three replications were broken. The branching stage occurred 33–46 days after sowing, followed by the flowering stage at 51–64 days after sowing, the pod formation stage at 70–82 days, the seed filling stage at 88–102 days, and the maturity stage at 100–126 days after sowing.

Since the focus on the study was WUE, we measured the indexes of evapotranspiration (ET), biomass (B) and yield (Y) during the entire growth of the crop. After first euphylla appeared, we began to weight each pot every other day for calculated evapotranspiration. Aboveground biomass (B) was measured at each growth stage. Cut two soybean plants in the pot with shear. The aboveground biomass were dried by oven at 80°C and wrote down the weight. Yield (Y) was measured at maturity. Weighted the naturally dried beans in each pot.” (Line 97 to 153 on page 3 to 4)

Point 5: You mentioned above that the total rainfall was more than 500 mm distributed from May to September covering quite the growing period. So, why you are talking about water stress? And is this the so called "Water stress"?

Response 5: We are sorry for making you feel confused. We have revised it in the manuscript. “Water use efficiency (WUE) is often considered an important determinant of yield under stress and even as a component of crop drought resistant. It can be expressed as follows:” (Line 176 to 177 on page 5)

In the manuscript, annual precipitation in northeast China with about 534 mm is average annual value for long term and rainfall seasonal distribution were uneven in this area. In our study, we found the critical stages of crop water requirement. If the rainfall is not as enough as crop water requirement at critical stages, water stress will happen.

Point 6: How you measured ET?

Response 6: We have added it in the manuscript.

“After first euphylla appeared, we began to weight each pot every other day for calculated evapotranspiration.” (Line 149 to 150 on page 4)

“The formula for evapotranspiration calculation is as follows:

   (2)

where i is one of the four soil water content levels,  is the total weight of a pot filled with soil at the ith designated soil water content level (g),  is the weight of a pot that record every other day (g), R is the radius of a pot with the value of 8 (cm), d is the interval days for measured with the value of 2 (day).” (Line 180 to 185 on page 5)

Point 7: Write again the meaning of the symbols to facilitate the readers.

Response 7: Thanks for your suggestion. We have added meaning of the symbols in the manuscript. “For biomass and yield, the optimal water content was 80%FC for lightly eroded soil (L) and 100%FC for both moderately eroded soil (M) and severely eroded soil (S).” (Line 210 to 212 on page 5)

Point 8: L and M look as the same. It is true since they do not significantly differ. Organic matter content is almost the same.

Response 8: Very good question. You are right. L and M look as the same organic matter content. But soil erosion have changed the mechanical composition, especially for sand contents which effected the water holding capacity. The differences of soil water indices were tested by us (Table a). Although organic matter content is almost the same, the water holding capacity were different.

Table a. The paired t-Test.

Paired Differences

t

df

Sig. (2-tailed)

Mean

Std. Deviation

Std. Error Mean

95% Confidence Interval of the difference

Lower

Upper

Pair 1

L(FC)-M(FC)

12.60250

7.47541

3.73770

.70746

24.49754

3.372

3

.043

Pair 2

L(WP)-M(WP)

4.18000

2.62226

1.17271

.92403

7.43597

3.564

4

.023

Pair 3

L(AWC)-M(AWC)

15.16000

2.77242

1.60066

8.27293

22.04707

9.471

2

.011

FC, field capacity; WP, wilting point; AWC, soil available water capacity.

For M, the contents of sand were higher than L, reduced soil’s effective capacity. The lower of the effective water content of the soil is, the more the soil needs to maintain the maximum effective water for crop growth. So the optimal water content was 100%FC for M, while 80%FC for L. Therefore, when the water supply was reduced, the water stress in M occurred, but L did not water stress occurred. So, when the water content was 80%FC, 60%FC and 40%FC, L and M were significantly differ.

Point 9: Again L and M gave the same results in figure 2

Response 9: Results in figure 2, were the optimal water conditions that under the condition of sufficient water supply. In the condition of sufficient water supply, the stress factors for crop growth were fertilities. As you said that the organic matters were similar in the L and M. So the results for L and M were the same.

Point 10: L and M differ only WUEB at 80% and 60% FC

Response 10: . Soil erosion have changed the mechanical composition, especially for sand contents which effected the water holding capacity. Under water stress, biomass and evapotranspiration were difference for L and M at 80% and 60% FC in figure 1(â… , â…¡). Therefore, they were different at 80%FC and 60%FC in WUEB.

Point 11: The difference in SOM looks too small to cause this reduce in WHC

Response 11: Very good question. Soil erosion have changed the mechanical composition, especially for sand contents which effected the water holding capacity. Although organic matter content is almost the same, the water holding capacity were different in Table a. In this study, we were focus on the water use efficiency on soybean, few works on the effect of SOM. Your questions provide a new research direction for us in the future.

Table a. The paired t-Test.

Paired Differences

t

df

Sig. (2-tailed)

Mean

Std. Deviation

Std. Error Mean

95% Confidence Interval of the difference

Lower

Upper

Pair 1

L(FC)-M(FC)

12.60250

7.47541

3.73770

.70746

24.49754

3.372

3

.043

Pair 2

L(WP)-M(WP)

4.18000

2.62226

1.17271

.92403

7.43597

3.564

4

.023

Pair 3

L(AWC)-M(AWC)

15.16000

2.77242

1.60066

8.27293

22.04707

9.471

2

.011

FC, field capacity; WP, wilting point; AWC, soil available water capacity.

Point 12: Give the names of the authors in all similar cases.

Response 12: Thanks for your suggestion. We have revised them in the manuscript.

Point 13: Which was the SOM reduction in this study?

Response 13: We are sorry for making you feel confused. “In this study, the organic matter content decreased from 5.85 to 1.96 % from L to S.” (Line 337 to 338 on page 11)

Point 14: This difference was

Response 14: We are sorry for making you feel confused. We have revised it in the manuscript. “For M and S, the contents of silt were 24.95% and 15.35%, and the clay contents were 31.71% and 16.52%, respectively, however, the sand contents for both of them were 43.34% and 68.13%, respectively.” (Line 334 to 336 on page 11)

Point 15: The soil erosion rate of this study not in general soil erosion.

Response 15: Thanks for your suggestion. We have revised it in the manuscript. “In this study, soil erosion had no significant effect on the response of soybean biomass and yield to water stress, and the corresponding coefficient of yield was 1.10–1.12.” (Line 345 to 346 on page 11)

Reviewer 2 Report

General Comment

The paper is well designed and the results are interesting.  The authors worked with soils with three different states of erosion and obtained the relationship between soil erosion, water-use and crop productivity. I suggest that the authors should clearly at the end and  in the abstract state the major implications of the study.  What should we bear in mind why planting crops in eroded soils?  This information will make the paper richer. The authors should state how the initial soil properties shown in Table 1 were measured.  References to the methods will be sufficient The paper has several errors in tenses and some of the sentences are not complete.  I have pointed out some of them below:

Line 11. Start with:  Soil erosion cold change the effective storage of soil moisture and affect water use efficiency (WUE).

Line 18:  Define the symbols e.g.  L, M, S, Y, B.  Without there meanings, it will be difficult for the reader to understand the abstract.  Alternatively, you can remove the symbols and generalise the results as much as possible.

Line 22:  What is SF?

Comment:  Add a last sentence to the abstract which should tell us the major finding or implication of the study.  This statement should state what could motivate the reader to continue reading this manuscript.

Lines 37 and 38:  Write:  Third, since soil erosion increases runoff, reduces water infiltration [23] as well as soil's effective capacity and influences the soil water storage eventually [24 - 33], poor   ....

Line 65: extraordinarily

Line 89: Should be:  damage of the soil physical properties and weaken the soil infiltration capacity [49, 53]

Line 91:  Should be: is sown in early May and reaped in late September

Line 93: soils not soil

Line 97:  How were the soil properties measured?

Line 101: to and not for

Line 106: closed not close

Line 106: that they were not

Line 107: To not For

Lines 124 and 125: Since the focus of the study was WUE, we measured ......ET was measured

Line 126: was not were; was not were

Line 165: soils not soil

Line 166: significant difference between L and M; from not for

Line 172: Why use best? Say higher or lower.  Avoid the use of best in technical writing as much as you can.

Line 214: both of

Line 283: Should be: Yield was best in M under optimal moisture which might be caused by soil texture changes after erosion

Line 290: Not a complete sentence:  Thus, with no nutrient limitations.  Please recap

Lines 292 to 294:  Sentence looks difficult to understand

Line 317:  Check the meaning of the first sentence

Line 323: focused on not focused

Line 347: State: There was no significant difference among the three eroded soils.

Author Response

Response to Reviewer2 Comments

The paper is well designed and the results are interesting. The authors worked with soils with three different states of erosion and obtained the relationship between soil erosion, water-use and crop productivity. I suggest that the authors should clearly at the end and in the abstract state the major implications of the study. What should we bear in mind why planting crops in eroded soils? This information will make the paper richer. The authors should state how the initial soil properties shown in Table 1 were measured. References to the methods will be sufficient. The paper has several errors in tenses and some of the sentences are not complete. I have pointed out some of them below:

Point 1: Line 11. Start with:  Soil erosion cold change the effective storage of soil moisture and affect water use efficiency (WUE).

Response 1: Thanks for your suggestion. We have revised it in the manuscript.

Point 2: Line 18:  Define the symbols e.g.  L, M, S, Y, B.  Without there meanings, it will be difficult for the reader to understand the abstract.  Alternatively, you can remove the symbols and generalise the results as much as possible.

Response 2: Thanks for your suggestion. We have revised it in the manuscript.

Point 3: Line 22:  What is SF?

Response 3: SF is water stress coefficient which was defined from line 198 to line 204 in the manuscript. We have revised the sentence “The stress coefficient (SF) values of the three eroded soils were not significantly different.” (Line 23 to line 24 on page 1)

Point 4: Add a last sentence to the abstract which should tell us the major finding or implication of the study. This statement should state what could motivate the reader to continue reading this manuscript.

Response 4: Thanks for your suggestion. We have added the sentences to the abstract which told readers the major finding. “This study was clarified indicated the change in soybean growth with respect to the water response caused by soil erosion and provided a scientific basis and data for the reasonable utilization of black soil with different erosion intensities. The results also provided important parameters for the growth of simulated crops.” (Line 27 to 31 on page 1)

Point 5: Lines 37 and 38:  Write:  Third, since soil erosion increases runoff, reduces water infiltration [23] as well as soil's effective capacity and influences the soil water storage eventually [24 - 33], poor   ....

Response 5: Thanks for your suggestion. We have revised it in the manuscript.

Point 6: Line 65: extraordinarily

Response 6: Thanks for your suggestion. We have revised it in the manuscript.

Point 7: Line 89: Should be:  damage of the soil physical properties and weaken the soil infiltration capacity [49, 53]

Response 7: Thanks for your suggestion. We have revised it in the manuscript.

Point 8: Line 91:  Should be: is sown in early May and reaped in late September

Response 8: Thanks for your suggestion. We have revised it in the manuscript.

Point 9: Line 93: soils not soil

Response 9: Thanks for your suggestion. We have revised it in the manuscript.

Point 10: Line 97:  How were the soil properties measured?

Response 10: Thanks for your suggestions. We have added the section of 2.3. Soil physicochemical properties. In that, the soil properties measured were clear.

“2.3. Soil physicochemical properties

Soil samples were collected in layers of 0-15cm and 15-30cm, separately at each sampling point. The mechanical composition, organic matter, bulk density, wilting points, and available nitrogen, phosphorus and potassium were measured for each sample. (1) Soil samples taken from each depth were mixed together. The natural air dried samples were sieved through 2 mm and used the pipette method to determine mechanical composition, and sieved through 0.149 mm and used the potassium dichromate volumetric method to determine soil organic matter. (2) The undisturbed soil was collected using 100 cm3 cutting rings and 3 repetitions of each depth. Total of 18 cutting rings were collected to determine soil saturated water content and bulk density. (3) Wilting points were measured with pressure membrane apparatus. The undisturbed soil was collected using a rigid PVC ring with a diameter of 4 cm and height of 1cm with 6 repetitions each, and 36 soil samples were collected. (4) Available nitrogen, phosphorus and potassium were measured with alkaline hydrolysis-diffusion method, sodium bicarbonate extraction technology and ammonium acetate extraction -flame photometer, respectively.” (Line 154 to 167 on page 4)

Point 11: Line 101: to and not for

Response 11: Thanks for your suggestion. We have revised it in the manuscript.

Point 12: Line 106: closed not close

Response 12: Thanks for your suggestion. We have revised it in the manuscript.

Point 13: Line 106: that they were not

Response 13: Thanks for your suggestion. We have revised it in the manuscript.

Point 14: Line 107: To not For

Response 14: Thanks for your suggestion. We have revised it in the manuscript.

Point 15: Lines 124 and 125: Since the focus of the study was WUE, we measured ......ET was measured

Response 15: Thanks for your suggestion. We have revised it in the manuscript.

Point 16: Line 126: was not were; was not were

Response 16: Thanks for your suggestion. We have revised it in the manuscript.

Point 17: Line 165: soils not soil

Response 17: Thanks for your suggestion. We have revised it in the manuscript.

Point 18: Line 166: significant difference between L and M; from not for

Response 18: Thanks for your suggestion. We have revised it in the manuscript.

Point 19: Line 172: Why use best? Say higher or lower. Avoid the use of best in technical writing as much as you can.

Response 19: Thanks for your suggestion. We have revised it in the manuscript.

Point 20: Line 214: both of

Response 20: Thanks for your suggestion. We have revised it in the manuscript.

Point 21: Line 283: Should be: Yield was best in M under optimal moisture which might be caused by soil texture changes after erosion

Response 21: Thanks for your suggestion. We have revised it in the manuscript.

Point 22: Line 290: Not a complete sentence: Thus, with no nutrient limitations.  Please recap

Response 22: Thanks for your suggestion. We have revised it in the manuscript.

Point 23: Lines 292 to 294: Sentence looks difficult to understand

Response 23: Thanks for your suggestion. We have revised it in the manuscript. “Many studies have confirmed that soil with lower bulk density and higher clay content could lead to the restriction of root growth and further reduced the yield [64-70]”. (Line 341 to line 343 on page 11)

Point 24: Line 317:  Check the meaning of the first sentence

Response 24: Thanks for your suggestion. We have revised it in the manuscript. “It is important to recognize these critical growth stages of crop water requirement.” (Line 367 on page 11)

Point 25: Line 323: focused on not focused

Response 25: Thanks for your suggestion. We have revised it in the manuscript.

Point 26: Line 347: State: There was no significant difference among the three eroded soils.

Response 26: Thanks for your suggestion. We have revised it in the manuscript.

Round 2

Reviewer 1 Report

After making the changes suggested, I think the paper now is acceptable for publication.